# Determinants of long-term opioid use in hospitalized patients

**Siyana Kurteva**[1,2,3]*, **Michal Abrahamowicz**[1], **Daniala Weir**[4], **Tara Gomes**[5,6,7], **Robyn Tamblyn**[1,2,8,9]

**1** Department of Epidemiology and Biostatistics, McGill University, Montreal, Canada, **2** Clinical and Health Informatics Research Group, McGill University, Montreal, Canada, **3** Science, Aetion, Inc., Barcelona, Spain, **4** Division of Pharmacoepidemiology and Clinical Pharmacology, Department of Pharmaceutical Sciences, Utrecht University, Utrecht, Netherlands, **5** Institute of Health Policy Management and Evaluation, Toronto, Canada, **6** Li Ka Shing Knowledge Institute, St. Michael's Hospital, Toronto, Canada, **7** ICES, Toronto, Canada, **8** Department of Medicine, McGill University Health Center, Montreal, Canada, **9** McGill University Health Centre, Montreal, Canada

* siyana.kurteva@mail.mcgill.ca

**Data Availability Statement:** We will not be able to share the governmental provincial administrative (RAMQ) data used in these analyses publicly. Our Research Ethics Board (REB) approval for this project limits data access to Robyn Tamblyn's

## Abstract

### Background

Long-term opioid use is an increasingly important problem related to the ongoing opioid epidemic. The purpose of this study was to identify patient, hospitalization and system-level determinants of long term opioid therapy (LTOT) among patients recently discharged from hospital.

### Design

To be eligible for this study, patient needed to have filled at least one opioid prescription three-months post-discharge. We retrieved data from the provincial health insurance agency to measure medical service and prescription drug use in the year prior to and after hospitalization. A multivariable Cox Proportional Hazards model was utilized to determine factors associated with time to the first LTOT occurrence, defined as time-varying cumulative opioid duration of $\geq$ 60 days.

### Results

Overall, 22.4% of the 1,551 study patients were classified as LTOT, who had a mean age of 66.3 years (SD = 14.3). Having no drug copay status (adjusted hazard ratio (aHR) 1.91, 95% CI: 1.40–2.60), being a LTOT user before the index hospitalization (aHR 6.05, 95% CI: 4.22–8.68) or having history of benzodiazepine use (aHR 1.43, 95% CI: 1.12–1.83) were all associated with an increased likelihood of LTOT. Cardiothoracic surgical patients had a 40% lower LTOT risk (aHR 0.55, 95% CI: 0.31–0.96) as compared to medical patients. Initial opioid dispensation of > 90 milligram morphine equivalents (MME) was also associated with higher likelihood of LTOT (aHR 2.08, 95% CI: 1.17–3.69).

### Conclusions and relevance

Several patient-level characteristics associated with an increased risk of $\geq$ 60 days of cumulative opioid use. The results could be used to help identify patients who are at high-risk of

team. To provide access to other researchers, they would need to obtain permission from the Institut de la Statistique du Québec for their project that would use RAMQ data. More details regarding eligibility to request data can be found here: https://statistique.quebec.ca/research/#/a-propos/utilisation-guichet. To use the clinical data, they would a) have to apply for REB approval for the project, and request that a de-identified copy of the data be made available as consent was provided by patients for only Robyn Tamblyn (consent form appended). More information regarding access to data can be found here: https://statistique.quebec.ca/research/#/demarche/etape-par-etape. A link to submit a data request can be found here: https://statistique.quebec.ca/zone-chercheur/#/login.

**Funding:** The author(s) received no specific funding for this work.

**Competing interests:** The authors have declared that no competing interests exist.

continuing opioids beyond guideline recommendations and inform policies to curb excessive opioid prescribing.

## Introduction

LTOT has been associated with an increased risk of opioid-related adverse events [1–3]. While opioids may be appropriate for short-term treatment for pain, LTOT has not been shown to improve pain relief [2, 4, 5]. The Centers for Disease Control and Prevention (CDC) guideline defines long-term opioid therapy (LTOT) as use of opioids on most days for more than 3 months [2], but in a recent empirical study we have found that the risk of opioid-related adverse events does not increase considerably beyond 60 days (Appendix D.1.a in S1 File) [6]. We observed only moderate increases in risk above 60 days of cumulative use with no evidence of further impact of durations longer than 100 days of use. There is a gap in evidence related to determinants of risk of LTOT using this empirically- derived definition of 60 days.

Previous studies have reported that the most common predictors associated with an increased risk for LTOT included age [7–11], sex [7–11], arthritis [9, 10, 12], race [8, 9], the presence of chronic pain [8–10] and mental illnesses such as anxiety and depression [8, 11–13]. Increased LTOT risk was also observed with certain characteristics of the first opioid pre-scription such as opioid doses of >90MME and longer duration (days' supply) [8, 9, 12, 14]. However, the evidence is limited to the data elements available in administrative data, such as patient and prescription characteristics in the community, and these databases do not typically have data on medications used in hospital or discharge information [15].

Hospitalization itself may inadvertently be a risk factor to initiating opioids [8], and inade-quate communication of changes in medication at the time of hospital discharge is also a well-established problem [16]. As a result, community physicians may continue opioids started in the hospital, for acute pain relief, as they have no information about the treatment indication nor the expected duration of therapy [17, 18].

There is a large variation in prescribing of opioids between providers [19], which is unre-lated to patient characteristics. The association between organizational /in-hospital prescriber characteristics and their contribution to the initiation, maintenance or prevention of LTOT is poorly understood. In addition, the type of opioid prescribed at hospital discharge varies across different attending physicians and residents [20, 21]. Thus, physician-prescribing behavior for post-discharge analgesia may be associated with an increased risk of LTOT. In this study, in addition to patient and medical characteristics, we were able to incorporate infor-mation on provider characteristics to better understand their contribution to the development of post-discharge LTOT. The purpose of this study was a) to estimate the proportion of hospi-talized patients with LTOT in the one year after hospital discharge, and b) to identify modifi-able patient, prescriber and system-level risk factors for long-term prescription opioid use compared to episodic use.

## Methods

### Setting

We carried out a secondary analysis of a cluster-randomized trial on discharge medication reconciliation conducted at the McGill University Health Centre (MUHC) [22]. The MUHC is an over 1000-bed quaternary care teaching hospital in Montreal (Canada) that operates within the universal health care plan of the province of Quebec (RAMQ). This plan covers all

hospitalizations, and essential medical care for provincial residents. It also provides drug insurance for registrants 65 years of age and older, income security recipients, and those not insured through their employer (approximately 50% of the Quebec population). Ethics approval was provided by the MUHC Research Ethics Board. Privacy Commissioner approval was obtained to link clinical and administrative data from the Commission d'accès à l'information du Quebec.

### Participants

A prospective cohort of medical and surgical hospitalized patients discharged from the MUHC between October 2014 and November 2016 were followed 12 months' post-discharge. To be eligible for the original trial, patients had to be 18 years of age or older at admission, admitted from the community or transferred from another hospital, with at least one-year continuous provincial healthcare coverage prior to hospital admission. To be included in this study, patients needed to fill at least one opioid prescription during the 90 days following their hospital discharge.

### Data sources

Multiple data sources were assembled and linked to address the study objectives. For each patient, demographic, clinical, healthcare use and prescription data were retrieved from admission notes as well as provincial healthcare administrative databases (RAMQ medical services and prescription claims) in the year prior to and after the hospitalization, for which the patient was enrolled. Dates of admission/discharge, admitting/discharge unit, patient demographics, diagnoses at admission and discharge, major procedures (surgeries, treatment interventions), were retrieved from the MED-ECHO hospitalization database. Medications at admission, in-hospital as well as those prescribed at discharge were abstracted from the MUHC Data Warehouse.

### Study measurements and predictors of long-term opioid use

**Long Term Opioid Use (LTOT).** Opioid use in the one-year post-discharge period was ascertained using RAMQ pharmacy administrative claims (see Appendix A.2 in S1 File for detailed description of LTOT calculation). As definitions of LTOT vary in different studies, standardized and evidence-driven definitions are needed [15]. In our recent study, the estimated non-linear effect of cumulative opioid duration showed no further increases in risk of opioid-related adverse events beyond 2 months of use (Appendix D.1.a in S1 File) [6]. Thus, in this study we used this empirically-defined 60 days' threshold to define LTOT.

**Patient-related characteristics.** RAMQ drug programs and hospital charts were used to collected information on age at admission, sex and socioeconomic status. Pain disorders, cancer diagnosis, mental health diagnoses, and conditions associated with abuse were measured at the time of hospital admission and during the hospital stay as fixed-in-time covariates identified by using ICD-9 from medical service claims and ICD-10 codes from hospitalization data. Other co-existing illnesses were measured and adjusted for using the Charlson comorbidity index (CCI), using information collected during the one-year baseline period (see Appendix 5-B in S1 File for a full list of covariates included in the model).

**Drug and healthcare utilization.** Psychoactive drugs such as antidepressants, benzodiazepines, Z-drugs or antipsychotics, when used together with opioids, have been associated with an increased risk of opioid-dependency and rates of adverse events [12, 13, 23]. Using RAMQ medical services databases in the one year prior to admission, we measured previous number

of emergency department (ED) visits, hospitalizations as well as the distinct number of pre-scribing physicians.

**Medication use and hospitalization characteristics.** Opioid and non-opioid medications administered as part of the in-hospital pain regimen as well as other medications, which may increase patients' risk of LTOT were extracted from the hospital pharmacy system using corre-sponding ATC codes. For surgical patients, information on the type of surgery received (tho-racic vs upper-gastrointestinal) was retrieved from the MED-ECHO hospitalization database.

**Prescribing physician and system-level characteristics.** Information on the attending physician's gender, language, training status (resident, attending physician) and number of years since licensure were abstracted from the patient medical chart, the medication reconcili-ation software databases and the hospital data warehouse.

**Opioid discharge prescription and initial dispensation.** Treatment changes made to opioid medications from the community were evaluated by using patients' discharge prescrip-tions data in comparison to their community drug list. A categorical variable for whether a given opioid was stopped, continued or newly prescribed was derived. A binary variable for the presence of an opioid as part of the discharge pain regimen was constructed: patients with newly added or continued opioids were flagged as having an opioid prescription. In addition, dose, duration and type of initial opioid dispensed (e.g. oxycodone, hydromorphone) were extracted from the RAMQ pharmacy claims.

## Statistical analyses

Descriptive statistics were used to compare LTOT patients versus episodic users with respect to patient, provider and hospital unit characteristics. Main analyses relied on time-to-event methodology [24]. A multivariable Cox proportional hazards (PH) model was utilized to determine which factors were associated with the development of LTOT within the one year post-discharge period. Start of follow-up corresponded to the date of the first opioid dispensa-tion. End of follow-up corresponded to the day when the patient first met the criteria for LTOT, or to right censoring at the end of follow-up or death, whichever came first. Moreover, since a patient is considered exposed based on periods of medication possession, patients were temporarily censored during subsequent re-admissions. Since we explored independent asso-ciations of various factors potentially related to LTOT, all *a priori* selected covariates were retained in the model. In addition, since patients could have subsequent hospitalizations and ED visits that could influence their risk of LTOT, a time-varying variable for the cumulative number of past post-discharge hospitalizations and ED visits, updated during the follow-up period, was included to adjust for any changes made to patient's medications. For each covari-ate, the results were presented as adjusted hazard ratios (aHR), with 95% confidence intervals (CI). We tested the PH assumption, both globally and for each covariate in the multivariable Cox model, using the Grambsch and Therneau approach [25], incorporated in *Survival pack-age*. For covariates for which the PH hypothesis was violated (p<0.05), we relied on smooth residual plots to assess how the corresponding adjusted hazard ratio varied over the follow-up (Appendix E in S1 File) [26, 27].

## Sensitivity analyses

To account for the fact that patient medications and medical history would most likely change over the course of one year, we updated information on selected co-morbidities and repre-sented them as additional time-varying covariates. We also adjusted for a time-varying count of distinct prescribers, from discharge until a given day, as an indicator of fragmentation in care, which may be due to increased opioid seeking behavior [28, 29]. These analyses were

**Table 1. Comparison of average daily and starting opioid dose between episodic and long-term opioid users by number of cumulative and number of continuous days of use to define long term use.**

|  | Number of People | Time to Long-Term Use | Average Daily Dose (MME) | Average Starting Dose (MME) |
|---|---|---|---|---|
|  | N (%) | Mean (SD) | Mean (SD) | Mean (SD) |
| **Cumulative ≥60 days' supply of opioids** |  |  |  |  |
| Episodic Opioid Users | 1173 (77.6) | 343.0 (69.6) | 35.7 (27.4) | 33.8 (22.7) |
| Long-Term Opioid Users | 338 (22.4) | 115.5 (76.8) | 57.4 (76.5) | 42.0 (44.9) |

additionally adjusted for a time-varying indicator of being currently exposed to ≥2 opioid products. Moreover, to account for the fact that the association of interest may vary depending on previous opioid use, we stratified the analyses by (i) previous LTOT and (ii) new opioid users. Due to small sample sizes within each stratum, variable selection was necessary, and was based on a combination of substantive knowledge and backward selection. To account for the cluster randomized design of the original trial and, thus, to assess potential clustering, i.e. interdependence, of the outcomes within the hospital units, we re-estimated the main multi-variable Cox model while adding unit-specific frailties, i.e, random intercepts, using R. Appendix F in S1 File summarizes the power calculations.

All MSM Cox PH models were implemented with SAS version 9.4 (SAS Institute, Cary, NC). All non-linear relationships were tested using customized programs in R [30].

## Results

Overall, 1511 patients were discharged alive from study units and filled an opioid prescription within 3 months' post-discharge. The proportion of patients who went on to become LTOT by accumulating more than 60 days' supply of opioids was 22.4% (n = 338) (Table 1), for the incidence rate of 26.8 (95% CI: 24.0–29.9) per 100 person-years. Among those 338 patients, the mean time to LTOT was 115.5 (SD = 76.8) days.

Fig 1 shows the breakdown of patients with respect to their previous history of opioid use, opioid administration during the hospitalization and the receipt of an opioid prescription at

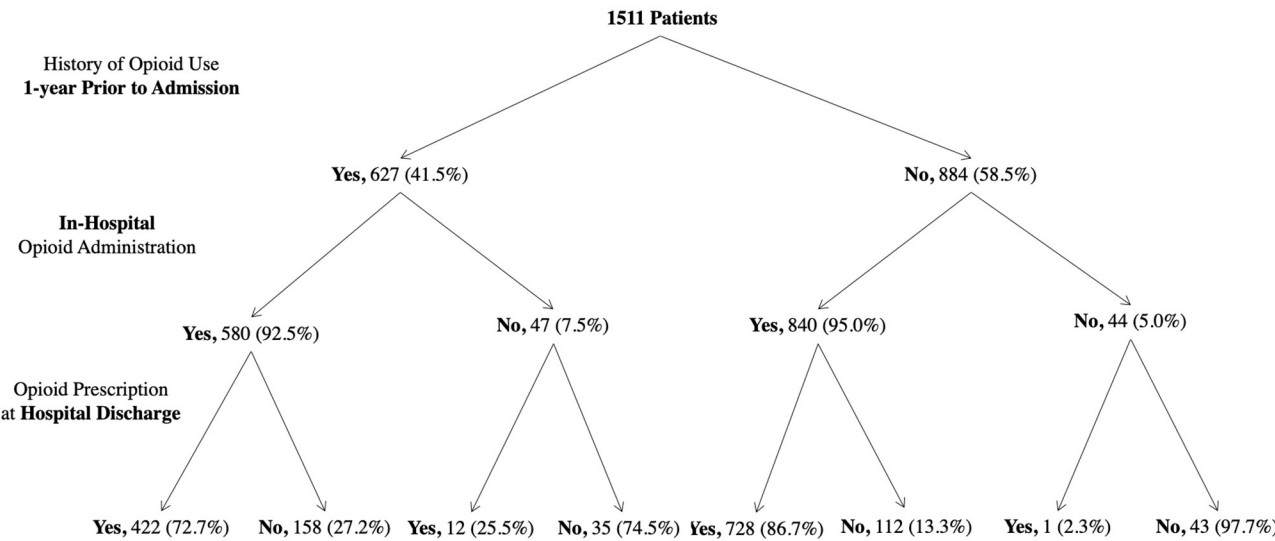

**Fig 1. Flowchart of patients' history of opioid use prior to admission, during the hospitalization, at hospital discharge and 3 month's post-discharge.**

hospital discharge. (Fig 1) Among patients who filled at least one opioid dispensation three months' post-discharge, more than half were opioid-naïve with no history of documented opioid use one year prior to their admission (n = 884, 58.5%).

The average age for LTOT patients was 66.3 (SD = 14.3) and more than half were female (179, 52.9%) (Table 2). LTOT patients were less likely to receive an opioid prescription at discharge, and more likely to have more than a 7-day supply on their first dispensation post-discharge (Table 3).

In multivariable analysis (Table 4), patients who had no copayment status had almost twice higher hazard of becoming LTOT users as compared to patients who had a 'full' copay drug insurance status (aHR 1.91, 95% CI: 1.40–2.60). As expected, patients who were previous LTOT users were several times more likely to also meet the criteria for LTOT in one-year post-discharge (aHR: 6.05, 95% CI: 4.22–8.68). History of benzodiazepine use (aHR: 1.43, 95% CI: 1.12–1.83), having a higher CCI (aHR: 1.77, 95% CI: 1.06–2.98)) and a starting daily opioid dose of >90 MME (aHR: 2.08, 95% CI: 1.17–3.69) were all independently associated with increased likelihood of becoming a LTOT user. Having undergone cardiothoracic surgery, as compared to internal medicine patients, was associated a 45% lower risk of LTOT in the post-discharge period (aHR 0.55, 95% CI: 0.31–0.96).

The PH hypothesis was rejected by the global test (p = 0.0046 and for four covariates (p<0.05 for each), for which Appendix E in S1 File describes how the corresponding aHR's varied with increasing follow-up.

Results from sensitivity analyses that adjusted for selected time-varying characteristics were similar, with a few exceptions. First, we found an association between recent cancer diagnoses and the risk of LTOT, showing an increased risk (aHR: 1.39, 95% CI: 1.07–1.81) (Appendix C.1 in S1 File). Second, after having adjusted for a time-varying count of distinct prescribing physicians and an indicator for using ≥2 opioid products, there was a 34% increased risk of LTOT use associated with having an initial days' supply of >7 days (aHR: 1.34, 95% CI: 1.05–1.72). Finally, having between 2 and 3 prescribing physicians led to more than doubling in the risk of becoming a LTOT user, relative to patients with only 1 prescriber (aHR: 2.43, 95% CI: 1.85–3.19) (Appendix C.2 in S1 File). In stratified analyses, out of all opioid-naïve patients (n = 1050), 117 (11.1%) went on to become LTOT users. On the other hand, among patients who were LTOT users in the year prior to admission (n = 192), only 68 (35.4%) were LTOT during the follow-up period. The risk of LTOT was 54% higher among patients with a history of pain syndromes compared to patients with no history (aHR: 1.54, 95% CI: 1.01–2.23) (Appendix C.4 in S1 File). The risk of LTOT was more than doubling when initial days' supply post-hospitalization exceeded >7 (aHR: 2.62, 95% CI: 1.52–4.53). There was no evidence of the clustering of the outcomes within individual hospital units (p = 0.96 for the frailty test).

## Discussion

Our study showed that 22.4% of hospitalized patients were characterized as LTOT users. There was an increased risk of LTOT among patients with no drug copay status, history of opioid use, history of benzodiazepine use, higher comorbidity index and higher starting daily dose in the first opioid dispensation post-discharge, whereas, surgical compared to medical patients had a decreased risk of LTOT.

Other studies have also confirmed that previous opioid use leads to greater risks of developing LTOT [7, 9–11, 13]. Previous research, which used a 90-day definition of LTOT, also found that mental health diagnoses, history of pain diagnoses and benzodiazepine use to be associated with an increased risk of LTOT [7–9]. In our study, these patients' characteristics were also associated with an increased risk, albeit some of these associations were non-

**Table 2. Characteristics of long term opioid users' vs episodic opioid users based on cumulative opioid ≥60 days' supply of opioids to define long-term use.**

| | Long-term Opioid Users | Episodic Opioid Users |
|---|---|---|
| | N = 338 (22.4%) | N = 1173 (77.6%) |
| *Patient Demographics* | | |
| | *N (%)* | *N (%)* |
| Age | | |
| Mean (SD) | 66.3 (14.3) | 67.0 (13.0) |
| ≤64 | 128 (37.9) | 360 (30.7) |
| >64 | 210 (62.1) | 813 (69.3) |
| Sex | | |
| Female | 179 (52.9) | 449 (38.3) |
| Male | 159 (47.0) | 724 (61.7) |
| Drug copay status | | |
| None | 111 (32.8) | 176 (15.0) |
| Partial | 76 (22.5) | 249 (21.2) |
| Full | 151 (44.7) | 748 (63.8) |
| *Healthcare and Medication Use: One Year Before Admission* | | |
| | *N (%)* | *N (%)* |
| Emergency department visits/Hospitalizations | 98 (29.0) | 406 (34.6) |
| Opioid use | | |
| No use | 125 (36.9) | 935 (79.1) |
| Episodic use (1–60 days) | 72 (21.3) | 187 (15.9) |
| Long-term opioid use (≥ 60 days) | 141 (41.7) | 51 (4.3) |
| Benzodiazepine use | 178 (52.7) | 333 (28.4) |
| Antidepressant use | 140 (41.4) | 201 (17.1) |
| Non-opioid pain medications use | 198 (58.6) | 355 (30.3) |
| *Comorbidities* | | |
| Mental illness/Substance & alcohol abuse | 80 (23.7) | 153 (13.0) |
| Charlson Comorbidity Index | | |
| 0 | 24 (7.1) | 195 (16.6) |
| 1–2 | 97 (28.7) | 438 (37.3) |
| ≥3 | 217 (64.2) | 540 (46.0) |
| *Characteristics Measured During the Hospitalization* | | |
| **Medications Administered** | | |
| Opioids | 300 (88.8) | 1120 (95.5) |
| Non-opioid pain medications | 226 (66.9) | 885 (75.4) |
| **Type of Surgery Received** | | |
| No surgery | 207 (61.2) | 297 (25.3) |
| Cardiothoracic | 45 (13.3) | 469 (39.9) |
| Gastrointestinal | 7 (2.1) | 45 (3.8) |
| Thoracic | 69 (20.4) | 308 (26.3) |
| Unrelated | 10 (3.0) | 54 (4.6) |
| **Admission to the ICU** | 29 (8.6) | 169 (14.4) |
| *Hospital Discharge Prescription* | | |
| **Pain Regimen** | | |
| Opioid prescription coming from the in-hospital prescriber | 222 (65.7) | 967 (82.4) |
| *Treatment Indication* | | |
| Surgery | 131 (39.8) | 876 (74.7) |

(*Continued*)

**Table 2.** (Continued)

| | Long-term Opioid Users | Episodic Opioid Users |
|---|---|---|
| | N = 338 (22.4%) | N = 1173 (77.6%) |
| Cancer | 268 (79.3) | 966 (82.4) |
| Pain Syndromes | 222 (65.7) | 735 (62.7) |
| *System Level Characteristics* | | |
| **Years of Practice** | | |
| 0–20 | 104 (30.5) | 229 (19.5) |
| 20–40 | 164 (48.5) | 808 (68.9) |
| >40 | 70 (20.7) | 136 (11.6) |
| **Sex** | | |
| Male | 247 (73.3) | 1046 (89.8) |
| Female | 90 (26.7) | 117 (10.1) |
| **Pharmacist on the Discharging Unit** | 227 (67.2) | 674 (57.5) |
| **Discharge Prescription Signed By** | | |
| Attending physician | 93 (27.5) | 217 (18.5) |
| Resident | 245 (72.5) | 956 (81.5) |
| **Hospital Discharge Destination** | | |
| Home | 325 (96.2) | 1153 (98.3) |
| Long-term care facility | 13 (3.8) | 20 (1.7) |

**Note**: 174 people died during the follow-up, which is one year since their first opioid dispensation within 3 months' post-discharge. These patients were censored at the time of death.

significant. Having the first opioid prescription written by an in-hospital prescriber was not associated with an increased risk of LTOT. In addition, it has been frequently argued that prescribing behavior and physicians' characteristics may contribute to the opioid epidemic [31, 32]. In our study, however, none of the physicians' characteristics such as years of practice, sex or having a resident approve the discharge prescription were associated with greater risks of LTOT post-discharge. One study found that LTOT patients of residents were more likely to

**Table 3. Characteristics of the first opioid dispensation within 90 days' post-discharge.**

| | Long-term Opioid Users | Episodic Opioid Users |
|---|---|---|
| | N = 338 (22.4%) | N = 1173 (77.6%) |
| **Type of Opioid Dispensed** | | |
| Codeine | 21 (6.2) | 26 (2.2) |
| Hydromorphone | 138 (40.8) | 281 (23.9) |
| Morphine | 24 (7.1) | 44 (3.8) |
| Oxycodone | 136 (40.2) | 816 (69.6) |
| Fentanyl | 18 (5.3) | 4 (0.3) |
| **MME Dose** | | |
| ≤20 | 101 (29.9) | 316 (26.9) |
| 20–50 | 169 (50.0) | 707 (60.3) |
| 50–90 | 44 (13.0) | 142 (12.2) |
| >90 | 24 (7.1) | 8 (0.7) |
| **Days' Supply** | | |
| ≤7 | 123 (36.4) | 660 (56.3) |
| >7 | 215 (63.6) | 513 (43.7) |

**Table 4. The association between patient, medication and system-level characteristics and time to long term use within the one-year post-discharge.**

| | Hazard Ratio | 95% CI |
|---|---|---|
| *Patient Characteristics* | | |
| **Age** | | |
| ≤64 | Reference | Reference |
| >64 | 1.19 | 0.88–1.60 |
| **Sex** | | |
| Male | Reference | Reference |
| Female | 1.22 | 0.97–1.55 |
| **Drug copay status** | | |
| Full | Reference | Reference |
| Partial | 1.12 | 0.82–1.52 |
| None | 1.91 | 1.40–2.60 |
| *Healthcare and Medication Use: One Year Before Admission* | | |
| **Emergency department visits/hospitalizations** | | |
| 0 | Reference | Reference |
| ≥1 | 0.93 | 0.72–1.21 |
| **Opioid use** | | |
| No use | Reference | Reference |
| Episodic use (1–60 days) | 1.94 | 1.43–2.69 |
| Long-term opioid use (≥ 60 days) | 6.05 | 4.22–8.68 |
| **Benzodiazepine use** | | |
| No use | Reference | Reference |
| Use | 1.43 | 1.12–1.83 |
| **Antidepressant use** | | |
| No use | Reference | Reference |
| Use | 1.20 | 0.92–1.58 |
| **Non-opioid medications use** | | |
| No use | Reference | Reference |
| Use | 1.21 | 0.92–1.57 |
| *Comorbidities* | | |
| **Mental illness/Substance & alcohol use disorder** | | |
| No | Reference | Reference |
| Yes | 1.04 | 0.78–1.39 |
| **Charlson Comorbidity Index** | | |
| 0 | Reference | Reference |
| 1–2 | 1.54 | 0.94–2.51 |
| ≥3 | 1.77 | 1.06–2.98 |
| *Characteristics Measured During the Hospitalization* | | |
| **Medications Administered** | | |
| *Opioids* | | |
| No | Reference | Reference |
| Yes | 1.15 | 0.73–1.83 |
| *Non-opioid pain medications* | | |
| No use | Reference | Reference |
| Use | 0.60 | 0.37–1.02 |
| **Type of Surgery Received** | | |
| No surgery | Reference | Reference |

(*Continued*)

**Table 4.** (Continued)

| | Hazard Ratio | 95% CI |
|---|---|---|
| Cardiothoracic | 0.55 | 0.31–0.96 |
| Gastrointestinal | 0.81 | 0.34–1.93 |
| Thoracic | 0.88 | 0.53–1.47 |
| Unrelated | 0.64 | 0.30–1.39 |
| *Hospital Discharge Prescription* | | |
| **Pain Regimen** | | |
| *Opioid prescription coming from the in-hospital prescriber* | | |
| No | Reference | Reference |
| Yes | 0.87 | 0.67–1.14 |
| **Treatment Indication** | | |
| *Cancer* | | |
| No | Reference | Reference |
| Yes | 1.01 | 0.74–1.38 |
| *Pain Syndromes* | | |
| No | Reference | Reference |
| Yes | 1.23 | 0.96–1.59 |
| *System Level Characteristics* | | |
| **Attending Physician Characteristics** | | |
| *Years of Practice* | | |
| 0–20 | Reference | Reference |
| 20–40 | 1.13 | 0.83–1.54 |
| >40 | 1.21 | 0.83–1.77 |
| *Sex* | | |
| Male | Reference | Reference |
| Female | 0.99 | 0.71–1.39 |
| *Language* | | |
| English | Reference | Reference |
| French | 1.05 | 0.77–1.43 |
| **Discharge Prescription Signed By** | | |
| Attending physician | Reference | Reference |
| Resident | 0.95 | 0.73–1.25 |
| **Hospital Discharge Destination** | | |
| Home | Reference | Reference |
| Long-term care facility | 1.06 | 0.58–1.96 |
| *System Level Characteristics* | | |
| **Type of Opioid Dispensed** | | |
| Codeine | Reference | Reference |
| Hydromorphone | 0.78 | 0.47–1.31 |
| Morphine | 0.78 | 0.40–1.44 |
| Oxycodone | 0.68 | 0.39–1.17 |
| Fentanyl | 0.85 | 0.39–1.86 |
| **MME Dose** | | |
| $\leq 20$ | Reference | Reference |
| 20–50 | 1.00 | 0.75–1.35 |
| 50–90 | 0.99 | 0.65–1.52 |
| >90 | 2.08 | 1.17–3.69 |
| **Days' Supply** | | |

(*Continued*)

**Table 4.** (Continued)

| | Hazard Ratio | 95% CI |
|---|---|---|
| ≤7 | Reference | Reference |
| >7 | 1.21 | 0.95–1.56 |

**Note**: All results obtained using a Cox Proportional Hazards Model. All variables were included in the model.

receive early refills following their primary care clinic visits when compared to attending physicians' patients [20]. In our study, the lack of difference in risk of LTOT associated with the status of the in-hospital prescriber might be attributed to the fact that residents provided similar care with respect to opioid prescribing when compared to attending physicians because they are being trained and monitored by the same physicians. These findings show that in-hospital prescribers may have the opportunity to reduce the occurrence of LTOT by providing patients with adequate pain treatment strategies, without contributing to the development of LTOT.

We found that initial dose and days' supply of the opioid dispensation both lead to an increased likelihood of LTOT, with initial doses having a greater impact on the risk of LTOT than opioid days' supply. Previous research has also found that both initial dose and duration of opioid use were associated with an increased risk of LTOT [33–35]. A few of the studies, which examined the initial opioid prescription characteristics and the likelihood of LTOT, found initial duration to be associated with a higher likelihood of LTOT than initial doses [14, 34, 36]. We observed a similar trend in stratified analyses, when looking at the risk among previous LTOT users only. However, like most other studies, opioid dose was associated with greater increases in risk of LTOT for opioid-naïve patients [8, 12]. These findings suggest that selecting an optimal initial opioid duration may be more important than initial dose to reduce the risk of subsequent use, especially for previous users. Moreover, initial opioid exposure characteristics should be used to profile patients who might be at risk of transitioning into LTOT.

This study's strength is its ability to link data on medication use prior to admission, during the hospitalization and dispensations post-discharge. Using multiple data sources enhances the internal validity of the study by providing detailed covariate information. This allowed us to consider not only patient-, but also provider- and system-level predictors of LTOT. Accounting for healthcare services use during the follow-up and including measures such as number of unique prescribers allowed us to examine the effect of poor coordination of care, flare-ups and complications requiring walk-in and ED visits, on the risk of LTOT. The risk for LTOT has been considered as a central component of quality care assessment [37, 38]. Measuring and defining LTOT is key to understanding potential risk factors, monitoring prevalence and incidence of LTOT, and improving clinical practices [39]. Yet, in previous studies, arbitrarily-defined measures were used without addressing the appropriateness of their cut-offs. In this study, we apply a novel and evidence-driven approach to defining LTOT. In addition, definitions in previous studies only relied on having prescriptions filled during a specified window [9, 40–43]. In this study, we used a time-varying definition of opioid duration constructed based on days' supply and fill dates. This allowed us to account for gaps between prescriptions and overlapping dispensations, and capture more accurately consistent opioid use.

Some limitations of our work merit emphasis. First, we used prescription duration as recorded by the pharmacist, but since opioids are usually prescribed on a *prn* basis, exposure mismeasurement is possible. We did not capture actual opioid consumption. Nevertheless,

subsequent prescription fills are suggestive of patient's continual opioid consumption. Future research should use data from multiple healthcare systems to incorporates measures on patients' healthcare providers and practice environments, and replicate our findings in larger populations cohorts. There is also a possibility of residual confounding due to unmeasured confounders, such as pain severity. Our decision to include only patients with at least one opioid dispensation post-discharge, as well as to include time-varying measures of selected comorbities and patient's healthcare utilization, reduces concerns about potential bias due to confounding by indication.

## Conclusions

We found increases in the risk of LTOT with multiple patient-level characteristics. Quantifying factors associated with the development of LTOT post-discharge is an important step in identifying and targeting patients who need more frequent clinical vigilance and better pain treatment strategy.

## Supporting information

**S1 File.**
(DOCX)

## Acknowledgments

We thank Dr. Marie-Eve Beauchamp for providing statistical input and technical help with the analyses.

## Author Contributions

**Conceptualization:** Siyana Kurteva, Michal Abrahamowicz, Daniala Weir, Tara Gomes, Robyn Tamblyn.

**Formal analysis:** Siyana Kurteva.

**Investigation:** Siyana Kurteva, Robyn Tamblyn.

**Methodology:** Siyana Kurteva, Michal Abrahamowicz, Robyn Tamblyn.

**Project administration:** Siyana Kurteva, Robyn Tamblyn.

**Resources:** Robyn Tamblyn.

**Software:** Siyana Kurteva, Robyn Tamblyn.

**Supervision:** Michal Abrahamowicz, Robyn Tamblyn.

**Validation:** Robyn Tamblyn.

**Visualization:** Siyana Kurteva, Robyn Tamblyn.

**Writing – original draft:** Siyana Kurteva.

**Writing – review & editing:** Siyana Kurteva, Michal Abrahamowicz, Daniala Weir, Tara Gomes, Robyn Tamblyn.

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
