## [Decision Letter · Decision Letter 0]

9 Jun 2022

PONE-D-21-29780Determinants of Long-Term Opioid Use in Hospitalized PatientsPLOS ONE

Dear Dr. Kurteva,

Thank you for submitting your manuscript to PLOS ONE. After careful consideration, we feel that it has merit but does not fully meet PLOS ONE’s publication criteria as it currently stands. Therefore, we invite you to submit a revised version of the manuscript that addresses the points raised during the review process.

Please note that we have only been able to secure a single reviewer to assess your manuscript. We are issuing a decision on your manuscript at this point to prevent further delays in the evaluation of your manuscript. Please be aware that the editor who handles your revised manuscript might find it necessary to invite additional reviewers to assess this work once the revised manuscript is submitted. However, we will aim to proceed on the basis of this single review if possible. 

The manuscript has been evaluated by one reviewer, and his comments are available below. The reviewer has raised a number of concerns regarding the statistical analyses.

Could you please carefully revise the manuscript to address all comments raised?

We look forward to receiving your revised manuscript.

Kind regards,

Lorena Verduci

Staff Editor

PLOS ONE

Journal Requirements:

2. Please ensure you have included the registration number for the clinical trial referenced in the manuscript.

4. Please upload a copy of Figure 1, to which you refer in your text on page 7. If the figure is no longer to be included as part of the submission please remove all reference to it within the text.

5. We note you have included a table to which you do not refer in the text of your manuscript. Please ensure that you refer to Table 3 in your text; if accepted, production will need this reference to link the reader to the Table

:

Reviewers' comments:

Reviewer's Responses to Questions

**Comments to the Author**

1. Is the manuscript technically sound, and do the data support the conclusions?

Reviewer #1: Partly

2. Has the statistical analysis been performed appropriately and rigorously? 

Reviewer #1: No

3. Have the authors made all data underlying the findings in their manuscript fully available?

Reviewer #1: Yes

4. Is the manuscript presented in an intelligible fashion and written in standard English?

Reviewer #1: Yes

5. Review Comments to the Author

Reviewer #1: The focus of this manuscript is on performing survival analysis, specifically, Cox regression, on right-censored time-to-event data generated from a cluster-randomized trial on discharge medication reconcilliation. I have some questions on running the Cox modeling, which, when addressed, may lead to enhanced clarity in the presentation.

1. An important step in fitting the multivariable Cox model is to "assess the proportional hazards assumptions". This check actually leads to handling time-varying covariates (which the authors anyway did as part of sensitivity analysis). However, I would like to see the initial checks (p-values) related to PH assessment. See the link below; testing for propotional hazards is available in various software.

https://www.ncbi.nlm.nih.gov/pmc/articles/PMC6015946/

2. I understand this is a secondary analysis of a cluster-randomized trial. It would help the readers if a statement on sample size/power is again presented illustrating the desired effect sizes the authors wanted to achieve/start with, under the available $n$ (sample size).

3. The data is cluster-randomized; it is not clear where the "cluster" is? What are the clusters/units?

4. It is not clear if a "frailty" proportional hazards model was fitted, which accounts for the clustering issues in cluster-randomized trials. The writeup doesn't reveal so clearly; if not considered, why?

https://hal.archives-ouvertes.fr/hal-03185154/document

6. PLOS authors have the option to publish the peer review history of their article (what does this mean?). If published, this will include your full peer review and any attached files.

Reviewer #1: No

---

## [Author Response · Author response to Decision Letter 0]

15 Aug 2022

Please see file named “Response to Reviewers” for a detailed response.

---

## [Decision Letter · Decision Letter 1]

4 Sep 2022

PONE-D-21-29780R1Determinants of Long-Term Opioid Use in Hospitalized PatientsPLOS ONE

Dear Dr. Kurteva,

Thank you for submitting your manuscript to PLOS ONE. After careful consideration, we feel that it has merit but does not fully meet PLOS ONE’s publication criteria as it currently stands. Therefore, we invite you to submit a revised version of the manuscript that addresses the points raised during the review process.

We look forward to receiving your revised manuscript.

Kind regards,

Vijayaprakash Suppiah, PhD

Academic Editor

PLOS ONE

Reviewers' comments:

Reviewer's Responses to Questions

**Comments to the Author**

1. If the authors have adequately addressed your comments raised in a previous round of review and you feel that this manuscript is now acceptable for publication, you may indicate that here to bypass the “Comments to the Author” section, enter your conflict of interest statement in the “Confidential to Editor” section, and submit your "Accept" recommendation.

Reviewer #1: (No Response)

2. Is the manuscript technically sound, and do the data support the conclusions?

Reviewer #1: No

3. Has the statistical analysis been performed appropriately and rigorously? 

Reviewer #1: No

4. Have the authors made all data underlying the findings in their manuscript fully available?

Reviewer #1: No

5. Is the manuscript presented in an intelligible fashion and written in standard English?

Reviewer #1: No

6. Review Comments to the Author

Reviewer #1: If the PH assumption has been rejected, then what did the authors do as an alternative? They need to consider some other statistical models for fitting, such as the accelerated failure time, etc. The habit of just fitting a PH model, whenever it's possible, should be stopped. Better involve a statistician/biostatistician in the research team.

7. PLOS authors have the option to publish the peer review history of their article (what does this mean?). If published, this will include your full peer review and any attached files.

Reviewer #1: No

---

## [Author Response · Author response to Decision Letter 1]

20 Oct 2022

Responses to the Reviewer

Reviewer:

1. Is the manuscript presented in an intelligible fashion and written in standard English?

Reviewer #1: No

Response:

Unfortunately, the reviewer has not specified the particular problems in the English presentation of this manuscript. There are no spelling or grammatical errors. The background, methods results and discussion are presented in standard form. The first author has many prior publications in JAMA Network Open, American Journal of Epidemiology, Journal of Surgical Oncology, Value in Health, Medical Care, BMJ Open, Journal of the National Cancer Institute among other journals and this the first time that there has been any criticism about the quality of the English presentation.

Reviewer:

2. If the PH assumption has been rejected, then what did the authors do as an alternative? They need to consider some other statistical models for fitting, such as the accelerated failure time, etc. The habit of just fitting a PH model, whenever it's possible, should be stopped. Better involve a statistician/biostatistician in the research team.

Response:

Below we provide separate responses to each of the 3 (different, even if partly inter-related) issues raised by the Reviewer in the above comment:

2.1. Re: Need to account for the rejection of the PH assumption: 

As reported in our revised manuscript (page 8, paragraph 3 of the unmarked version) the PH assumption was rejected for only 4 among 36 covariates considered in our multivariable model, while the relationships of the 32 remaining covariates were consistent with the PH hypothesis, indicating that the corresponding adjusted Hazard Ratios (HR) did not vary systematically during the follow-up. In this situation, consistent with statistical literature 1-3, we have relied on a flexible time-dependent extension of the multivariable PH model, to simultaneously estimate (a) constant adjusted HRs for the majority of the covariates that meet the PH assumption, combined with (b) adjusted time-dependent HRs for each of the 3 covariates that violate this assumption. Due to space restrictions, in the “Results” section of the main (revised) manuscript, we were able to only briefly mention this issue (page 8, paragraph 3) and then to refer the readers to Appendix E (page 10-114) of Supplementary Materials, which describes in more detail how the time-dependent HRs for each of the four covariates that violate the PH hypothesis vary over with increasing follow-up time. This approach provides insights about the role of each covariate and is consistent with recent recommendations of a comprehensive review paper that focuses on state-of-the-art applications of intensity-based (i.e., hazard-based) modeling in survival analyses. 1A similar approach, based on flexible time-dependent extension of the multivariable PH model, was employed in several other publications in high-ranked epidemiological and medical journals over the past three decades 4-7, including several very recent papers 8-10 

2.2. Re: Pros and cons of using an alternative regression model.

The Reviewer has suggested we may consider an alternative regression model, and specifically has mentioned the accelerated failure time (AFT) model. Interestingly, our team is not only well familiar with the AFT model but, in addition, has made recent methodological developments in this area of statistical research. In fact, the 2nd author (MA) is the corresponding/senior author of 2 papers, very recently published in top-ranking biostatistical journals (Statistics in Medicine and SMMR) that both extend the existing AFT modeling methodology to allow more flexible modeling of, respectively, (i) baseline hazard 11 and (ii) covariate effects 12 (the 1st author of both papers is his PhD student). Thus, in principle, we have both the software and the skills necessary to re-analyze our data using a multivariable AFT model and assessing its underlying assumptions. However, there are several reasons while we have chosen not to change a posteriori the general strategy for modeling our data and, thus, to present the results of the flexible extension of the PH model (discussed in point 1 above). (1) The fact that a vast majority of time-to-event analyses reported in medical literature (including earlier papers on the associations of opioids use with different health outcomes 13-17 , and many recent papers in PLOS One 18-23 employ the Cox PH model, makes it preferable to rely on this ‘conventional’, well understood, model to enhance both the interpretability of our results and their comparability with other studies in this area. (2) We are not aware of any published extension of the AFT model, or any currently available software for fitting the AFT model, that could handle time-varying covariates, that are included in our multivariable analyses (please see section on study measurements in our manuscript, page 6-7). (3) As explained in point 1 above, our revised manuscript and Appendix E of Supplementary Materials presents results that accurately account for violations of the PH assumption by a few (four) among 36 covariates. (4) Both theoretical considerations (more details on request) and empirical evidence based on our recent analyses of both simulated and real-world data12 [2 recent Invited Talks by the 2nd author (MA) at major international biostatistical conferences24,25] suggest that most covariates which violate the PH assumption will also likely violate the AFT assumption, i.e. will have time-dependent time ratios, in addition to time-dependent HR’s. Thus, relying on the AFT model will likely not avoid the need to discuss more complex time-dependent covariate effects. (5) Conceptually, we believe that AFT model is especially relevant to model biological relationships between some clinical factors, or laboratory biomarkers, and time to the occurrence of clinical outcomes such as death or cancer recurrence (see e.g., the monograph by Cox and Oakes). In contrast, it seems less plausible than a given subject characteristic may proportionally “accelerate’ or ‘decelerate’ time to such events as reaching the threshold for long term opioid therapy, defined as at least 60 days of cumulative duration of opioid therapy, that reflects a complex interplay between patients’ need for pain control, their socio-demographic characteristics, and physician prescribing preferences. Thus, we believe that in our setting it is more plausible to describe the underlying associations based on the PH model, and its flexible extensions, and the resulting, possibly time-dependent, Hazard Ratio estimates (rather than the Time Ratios corresponding to the AFT model). 

1. Re: need to involve a statistician/biostatistician in the research team.

We agree with the Reviewer that the analytical challenges of our study require an involvement of a (bio-)statistician well versed in survival analysis methodology. In fact, the 2nd author (MA) conducts active research in developing new methods for flexible modelling of multivariable time-to-event data. In particular, he is the senior or first author of several papers, published in top-ranking biostatistical journals (e.g. Statistics in Medicine, SMMR, JASA, Biometrical Journal) dealing with flexible extensions of: either (i) the PH model 26-30 or, (ii) more recently, the AFT model11,12, as well as many invited talks on these topics at major international biostatistical conferences (please see point 2 above for two recent examples). A recent Statistics in Medicine tutorial on accounting for violation of the PH assumption and flexible modeling of time-dependent Hazard Ratios cites several of his papers in this area. 31 He is also a co-Chair of the Survival Analysis Topic Group of the International STRATOS Initiative (www.stratos-initiative.org), for STRengthening Analytical Thinking for Observational Studies, and is the co-author of the recent STRATOS guidance/overview paper on current issues and state-of-the-art methods for survival analysis. 1

In conclusion, we hope that the evidence and the arguments presented above adequately address the Reviewer’s comments. 

References:

1. Kragh Andersen P, Pohar Perme M, van Houwelingen HC, et al. Analysis of time-to-event for observational studies: Guidance to the use of intensity models. Statistics in medicine. Jan 15 2021;40(1):185-211. doi:10.1002/sim.8757

2. Hastie T, Tibshirani R. Varying-Coefficient Models. Journal of the Royal Statistical Society Series B (Methodological). 1993;55(4):757-796. 

3. Kooperberg C, Stone CJ, Truong YK. Hazard Regression. Journal of the American Statistical Association. 1995;90(429):78-94. doi:10.2307/2291132

4. Rachet B, Sasco AJ, Abrahamowicz M, Benyamine D. Prognostic factors for mortality in nasopharyngeal cancer: accounting for time-dependence of relative risks. Int J Epidemiol. Oct 1998;27(5):772-80. doi:10.1093/ije/27.5.772

5. Quantin C, Abrahamowicz M, Moreau T, et al. Variation over time of the effects of prognostic factors in a population-based study of colon cancer: comparison of statistical models. American journal of epidemiology. Dec 1 1999;150(11):1188-200. doi:10.1093/oxfordjournals.aje.a009945

6. Côté R, Battista RN, Abrahamowicz M, Langlois Y, Bourque F, Mackey A. Lack of effect of aspirin in asymptomatic patients with carotid bruits and substantial carotid narrowing. The Asymptomatic Cervical Bruit Study Group. Annals of internal medicine. Nov 1 1995;123(9):649-55. doi:10.7326/0003-4819-123-9-199511010-00002

7. Gagnon B, Abrahamowicz M, Xiao Y, et al. Flexible modeling improves assessment of prognostic value of C-reactive protein in advanced non-small cell lung cancer. British journal of cancer. Mar 30 2010;102(7):1113-22. doi:10.1038/sj.bjc.6605603

8. Kurteva S, Abrahamowicz M, Beauchamp ME, Tamblyn R. Comparison of different modeling approaches for prescription opioid use and its association with adverse events. American Journal of Epidemiology (Accepted: August 17, 2022). 

9. Dimitris MC, Hutcheon JA, Platt RW, Abrahamowicz M, Beauchamp ME, Himes KP, Bodnar LM, Kaufman JS. Investigating the Shape and Strength of the Relationship Between Maternal Weight Gain and Gestational Age at Delivery in Twin and Singleton Pregnancies. American Journal of Epidemiology (Accepted: September 19, 2022)

10. Isidean SD, Wang Y, Mayrand MH, et al. Assessing the time dependence of prognostic values of cytology and human papillomavirus testing in cervical cancer screening. International journal of cancer. May 15 2019;144(10):2408-2418. doi:10.1002/ijc.31970

11. Pang M, Platt RW, Schuster T, Abrahamowicz M. Spline-based accelerated failure time model. Statistics in medicine. Jan 30 2021;40(2):481-497. doi:10.1002/sim.8786

12. Pang M, Platt RW, Schuster T, Abrahamowicz M. Flexible extension of the accelerated failure time model to account for nonlinear and time-dependent effects of covariates on the hazard. Statistical methods in medical research. Nov 2021;30(11):2526-2542. doi:10.1177/09622802211041759

13. Shah A, Hayes CJ, Martin BC. Factors Influencing Long-Term Opioid Use Among Opioid Naive Patients: An Examination of Initial Prescription Characteristics and Pain Etiologies. The journal of pain : official journal of the American Pain Society. Nov 2017;18(11):1374-1383. doi:10.1016/j.jpain.2017.06.010

14. Saunders KW, Dunn KM, Merrill JO, et al. Relationship of opioid use and dosage levels to fractures in older chronic pain patients. J Gen Intern Med. Apr 2010;25(4):310-5. doi:10.1007/s11606-009-1218-z

15. Waddy SP, Becerra AZ, Ward JB, et al. Concomitant Use of Gabapentinoids with Opioids Is Associated with Increased Mortality and Morbidity among Dialysis Patients. Am J Nephrol. 2020;51(6):424-432. doi:10.1159/000507725

16. Kurteva S, Abrahamowicz M, Gomes T, Tamblyn R. Association of Opioid Consumption Profiles After Hospitalization With Risk of Adverse Health Care Events. JAMA Netw Open. May 3 2021;4(5):e218782. doi:10.1001/jamanetworkopen.2021.8782

17. Miller M, Barber CW, Leatherman S, et al. Prescription opioid duration of action and the risk of unintentional overdose among patients receiving opioid therapy. JAMA Intern Med. Apr 2015;175(4):608-15. doi:10.1001/jamainternmed.2014.8071

18. Arora C, Kaur D, Naorem LD, Raghava GPS. Prognostic biomarkers for predicting papillary thyroid carcinoma patients at high risk using nine genes of apoptotic pathway. PloS one. 2021;16(11):e0259534. doi:10.1371/journal.pone.0259534

19. Boukebous B, Maillot C, Neouze A, et al. Excess mortality after hip fracture during COVID-19 pandemic: More about disruption, less about virulence-Lesson from a trauma center. PloS one. 2022;17(2):e0263680. doi:10.1371/journal.pone.0263680

20. Cohen AT, Sah J, Dhamane AD, et al. Effectiveness and safety of apixaban vs warfarin among venous thromboembolism patients at high-risk of bleeding. PloS one. 2022;17(9):e0274969. doi:10.1371/journal.pone.0274969

21. Fagbamigbe AF, Norrman E, Bergh C, Wennerholm UB, Petzold M. Comparison of the performances of survival analysis regression models for analysis of conception modes and risk of type-1 diabetes among 1985-2015 Swedish birth cohort. PloS one. 2021;16(6):e0253389. doi:10.1371/journal.pone.0253389

22. Serraino GF, Provenzano M, Jiritano F, et al. Risk factors for acute kidney injury and mortality in high risk patients undergoing cardiac surgery. PloS one. 2021;16(5):e0252209. doi:10.1371/journal.pone.0252209

23. Thammavaranucupt K, Phonyangnok B, Parapiboon W, et al. Metformin-associated lactic acidosis and factors associated with 30-day mortality. PloS one. 2022;17(8):e0273678. doi:10.1371/journal.pone.0273678

24. Abrahamowicz M. Invited Speaker. Selected challenges in multivariable time-to-event analyses. Invited Session "Various issues in multivariable model building". Royal Statistical Society (RSS) International Conference. Manchester, UK. September 6-9, 2021. 

25. Invited Speaker. Flexible extension of AFT model to account for non-linear effects and time-dependent effects of covariates on the hazard. Invited session on “Flexible extensions of the AFT model", 31th International Biometric Conference (IBC), Riga, Latvia, July 10-15, 2022. 

26. Wynant W, Abrahamowicz M. Flexible estimation of survival curves conditional on non-linear and time-dependent predictor effects. Statistics in medicine. Feb 20 2016;35(4):553-65. doi:10.1002/sim.6740

27. Wynant W, Abrahamowicz M. Impact of the model-building strategy on inference about nonlinear and time-dependent covariate effects in survival analysis. Statistics in medicine. Aug 30 2014;33(19):3318-37. doi:10.1002/sim.6178

28. Wang Y, Beauchamp ME, Abrahamowicz M. Nonlinear and time-dependent effects of sparsely measured continuous time-varying covariates in time-to-event analysis. Biom J. Mar 2020;62(2):492-515. doi:10.1002/bimj.201900042

29. Abrahamowicz M, MacKenzie TA. Joint estimation of time-dependent and non-linear effects of continuous covariates on survival. Statistics in medicine. Jan 30 2007;26(2):392-408. doi:10.1002/sim.2519

30. Abrahamowicz M, Mackenzie T, Esdaile JM. Time-Dependent Hazard Ratio: Modeling and Hypothesis Testing with Application in Lupus Nephritis. Journal of the American Statistical Association. 1996/12/01 1996;91(436):1432-1439. doi:10.1080/01621459.1996.10476711

31. Austin PC, Fang J, Lee DS. Using fractional polynomials and restricted cubic splines to model non-proportional hazards or time-varying covariate effects in the Cox regression model. Statistics in medicine. Feb 10 2022;41(3):612-624. doi:10.1002/sim.9259

---

## [Decision Letter · Decision Letter 2]

29 Nov 2022

Determinants of Long-Term Opioid Use in Hospitalized Patients

PONE-D-21-29780R2

Dear Dr. Kurteva,

We’re pleased to inform you that your manuscript has been judged scientifically suitable for publication and will be formally accepted for publication once it meets all outstanding technical requirements.

Kind regards,

Vijayaprakash Suppiah, PhD

Academic Editor

PLOS ONE

Reviewers' comments:

Reviewer's Responses to Questions

**Comments to the Author**

1. If the authors have adequately addressed your comments raised in a previous round of review and you feel that this manuscript is now acceptable for publication, you may indicate that here to bypass the “Comments to the Author” section, enter your conflict of interest statement in the “Confidential to Editor” section, and submit your "Accept" recommendation.

Reviewer #1: All comments have been addressed

2. Is the manuscript technically sound, and do the data support the conclusions?

Reviewer #1: (No Response)

3. Has the statistical analysis been performed appropriately and rigorously? 

Reviewer #1: (No Response)

4. Have the authors made all data underlying the findings in their manuscript fully available?

Reviewer #1: (No Response)

5. Is the manuscript presented in an intelligible fashion and written in standard English?

Reviewer #1: (No Response)

6. Review Comments to the Author

Reviewer #1: (No Response)

7. PLOS authors have the option to publish the peer review history of their article (what does this mean?). If published, this will include your full peer review and any attached files.

Reviewer #1: No

---

## [Editor Report · Acceptance letter]

6 Dec 2022

PONE-D-21-29780R2 

Determinants of Long-Term Opioid Use in Hospitalized Patients 

Dear Dr. Kurteva:

I'm pleased to inform you that your manuscript has been deemed suitable for publication in PLOS ONE. Congratulations! Your manuscript is now with our production department. 

Kind regards, 

on behalf of

Dr. Vijayaprakash Suppiah 

Academic Editor

PLOS ONE